# Distributed Data Placement and Content Delivery in Web Caches with Non-Metric Access Costs

## ABSTRACT

Motivated by applications in web caches and content delivery in peer-to-peer networks, we consider the non-metric data placement problem and develop distributed algorithms for computing or approximating its optimal solutions. In this problem, the goal is to store copies of the data points among a set of cache-capacitated servers to minimize overall data storage and clients' access costs. We first show that the non-metric data placement problem in which the access costs between servers can be arbitrary nonnegative numbers is inapproximable up to a logarithmic factor. We then provide a game-theoretic decomposition of the objective function and show that a natural type of Glauber dynamics in which servers update their cache contents with probability proportional to the utility they receive from caching those data will converge to an optimal global solution for a sufficiently large noise parameter. In particular, we establish the polynomial mixing time of the Glauber dynamics for a certain range of noise parameters. Such a game-theoretic decomposition not only provides a good performance guarantee in terms of content delivery but also allows the system to operate in a fully distributed manner, hence reducing the system's computational load and improving its robustness to system failures. Finally, we provide another auction-based distributed algorithm, which allows us to approximate the optimal global solution with a performance guarantee that depends on the ratio of the revenue vs. social welfare obtained from the underlying auction.

## KEYWORDS

Content delivery, web caches, distributed data placement, Glauber learning dynamics, potential games, LP duality.

**ACM Reference Format:**
Anonymous Author(s). 2023. Distributed Data Placement and Content Delivery in Web Caches with Non-Metric Access Costs. In . ACM, New York, NY, USA, 10 pages. https://doi.org/

Data placement is one of the fundamental allocation problems in storage-capable distributed systems, such as content delivery networks, web caches, peer-to-peer (P2P) networks, and mobile networks. In many such applications one aims to store limited copies of the data points among a set of cache-aided servers in order to improve data availability to the clients, enhance system reliability and fault tolerance (e.g., against adversarial attacks), and reduce the computational load at the local servers [16]. As an example, consider a movie-sharing P2P network among multiple servers [15]. Due to limited disk space, the movies can be stored either locally

© 2023 Association for Computing Machinery.
ACM ISBN ... $15.00
https://doi.org/

or obtained from other servers with some delay cost. The communication network shared by the servers determines the delay costs, where the farther the servers are from each other, the more delay costs they incur in obtaining access to each other. When a client connects to one of the local servers to download a certain movie, either the movie is available in the cache of that local server, in which case the client can get access to it without any delay. Otherwise, the client can access the movie through the closest server that has the requested movie in its cache, hence incurring a delay cost. Thus, the storing decisions at each local server affect everyone who uses this service, and a major question is on how to distribute the movies locally among the servers to minimize communication delays while respecting servers' capacity constraints. In particular, we often want the data distribution to be robust and reliable, meaning that if a server goes down or behaves selfishly, other servers can fulfill their clients' demands with a small delay cost.

The data placement problem can be viewed under a more general framework of resource allocation problems that looks at how to store copies of different data (resources) among a set of capacity-constrained servers (agents) to minimize the overall resource placement and access costs.[1] The placement cost captures the cost of allocating a particular resource to an agent (e.g., the storage cost on a server due to data incompatibility). On the other hand, the access costs measure the cost of getting access to resources from other agents (e.g., the data transmission delay between servers). However, most works on data placement problems assume that all agents fully comply with the centralized designed protocols. Nevertheless, in real-world data replication applications, the agents (e.g., servers, data providers/consumers) can belong to different stakeholders or administrative domains with different preferences and objectives [12, 13, 15, 16]. On the other hand, the access costs among agents could be a complex function of many parameters, such as agents' physical locations, available bandwidth, or the network congestion shared by them, and hence may not satisfy any simplifying assumptions. Therefore, our main goal in this work is to analyze the data placement problem from a distributed computation perspective and without any assumption on the access costs.

**Related Work:** The data placement problem has been extensively studied in the past literature. The optimal data placement on networks with a constant number of clients and arbitrary access costs was considered in [2], where a polynomial algorithm for computing the optimal allocation for uniform resource length was developed. The data placement problem was studied from a mechanism design perspective in [16]. There have been several efforts to obtain constant factor approximation algorithms for the *metric* data placement problem, starting with [5] and improved by [4, 20, 27]. The main idea behind most of these approximation algorithms is based on solving a natural LP relaxation of the problem and then rounding the solution using refined clustering, network flow, or

---

[1]We refer to servers as agents and data points as resources interchangeably.

iterative rounding. A generalization of these results to the so-called "matroid median problem" has been studied in [20, 27]; as a special case, it results in improved approximation algorithms for the metric data placement problem. We refer to [3, 28] for other heuristic approximation algorithms with or without theoretical performance guarantees. It was shown in [12] that in the case of homogeneous metric data placement when agents have identical request rates, a simple greedy algorithm could achieve an approximation factor of 3, hence improving the existing approximation factors that were given for the heterogeneous setting. Moreover, it was shown that the same algorithm results in a 3-competitive algorithm for the online version of the problem in which agents arrive adversarially over time and reveal their specifications. A different online variant of the data placement problem has also been studied in [10].

The data placement problem is also closely related to the uncapacitated facility location problem (UFLP) [19] and its variants [1, 18, 26], such as the $k$-median and matroid median problems [7, 9], in which the goal is to open a subset of facilities and assign each client to an open facility in order to minimize the total facility opening costs and clients' access costs. In fact, as we will show, the data placement problem is a more complicated version of the UFLP in which there are multiple facility types that are coupled through the cache constraints. In particular, one can show that by relaxing the cache constraints in the data placement problem using Lagrangian multipliers, the data placement problem can be decomposed into a sum of separable UFLPs. A heuristic approximation algorithm based on decomposing the data placement objective function using Lagrangian relaxation has been studied in [11].

This work is also related to the literature on distributed caching using auctions [6] and resource allocation games [14, 22–25]. One advantage of studying content delivery and web caching problems using distributed game-theoretic frameworks is improving resource availability and robustness against failures [8, 13]. An uncapacited distributed caching game was studied in [8], where the authors were able to characterize the set of Nash equilibrium points. However, the situation is more complex for content delivery systems with limited cache size, as the agents' caching decisions are more coupled such that there is no simple characterization of the equilibrium points at hand. The capacitated distributed caching games were studied in [12, 13, 15], where it was shown that the game admits a pure Nash equilibrium if the access costs are metric and the request rates are uniform across agents. Our work is different from the prior work as it provides a novel game-theoretic decomposition for the distributed caching problem with different request rates and general access costs.

**Contributions:** We consider the non-metric data placement problem and devise distributed algorithms to compute or approximate its global optimum solution. More specifically:

- We first show that the data placement problem with non-metric access costs is NP-hard to approximate within a factor better than $O(\log n)$, where $n$ is the number of agents.
- We then provide a novel potential game decomposition of the data placement problem that allows the agents to internalize the global cost through their own local cost functions. Such a decomposition aligns the changes in the global objective function with the agents' selfish behavior, allowing the system to operate in a fully distributed manner.

- We obtain a bound on the objective value of any pure Nash equilibrium (NE) in terms of the sensitivity to agents' cache contents. In general, the objective value of a NE could be far from an optimum value. However, we show that natural distributed Glauber dynamics will allow the agents to randomly update their resources to collectively achieve an optimum solution for a reasonably large noise parameter. Moreover, the Glauber dynamics mix quickly if the noise parameter is within a certain range, hence providing a clear tradeoff between the computation time and the quality of the suboptimal solution obtained from the dynamics.
- Finally, using the dual formulation of the data placement problem, we provide another distributed algorithm based on the first price auction, in which the resources bid on agents' cache capacities and provide a worst-case guarantee on the quality of the obtained solution.

Our results provide the first distributed computation algorithms for the non-metric data placement problem and provide new insights on how to trade-off between the computation cost versus the optimality of the achieved solution in a fully distributed setting.

**Notations:** We adopt the following notations throughout the paper. For a positive integer $n$, we let $[n] = \{1, 2, \ldots, n\}$. For a discrete set $X \subseteq [n]$ and $i \in X$, we often write $X - i$ and $X + i$ to denote $X \setminus \{i\}$ and $X \cup \{i\}$, respectively. For two probability distributions $\mu$ and $\nu$ supported over a finite set $A$, we let $\|\mu - \nu\|_{TV} = \frac{1}{2} \sum_{a \in A} |\mu(a) - \nu(a)|$ be the total variation distance between those distributions. To denote the mixing time of a Markov chain with transition probability matrix $P$, we use $t_{\min}(\epsilon) = \min\{t : d(t) < \epsilon\}$, where $d(t) = \sup_\mu \|\mu P^t - \pi\|_{TV}$ is the maximum total variation between the distribution of the Markov chain at time $t$ and its stationary distribution $\pi$. The $\ell_1$ norm of a vector $x$ is denoted by $\|x\|_1 = \sum |x_i|$. Given two vectors $x$ and $y$, we let $\rho(x, y)$ be the number of coordinates for which those two vectors differ. We let $\mathbf{1}$ and $\mathbf{0}$ be the vectors with all one entries and all zero entries, respectively. Finally, for a real number $a$, we let $(a)^+ = \max\{a, 0\}$.

## 1 PROBLEM FORMULATION AND PRELIMINARY RESULTS

Let us consider the data placement problem [5, 27] in which there are a set of $[n] = \{1, 2, \ldots, n\}$ agents and a set $[k] = \{1, 2, \ldots, k\}$ of unit size resource types. We assume that there are unlimited copies of each resource type. Agent $i \in [n]$ has a cache of size $u_i \in \mathbb{Z}_+$, and the cost of storing resource $\ell$ in its cache is given by $f_i^\ell \geq 0$. Moreover, we assume that agent $i \in [n]$ has a nonnegative demand/request rate $w_i^\ell \geq 0$ for resource $\ell$. Here, $w_i^\ell$ can be thought of as the rate of clients who refer to agent $i$ to request resource $\ell$ and is a measure of the popularity of resource $\ell$ at server $i$. In addition, we let $c_{ij} = c_{ji} \geq 0$ be the (symmetric) cost of getting access to agent $i$ from agent $j$, and $c_{ii} = 0 \ \forall i \in [n]$. The goal is to fill agents' caches with proper resources to minimize the overall placement and access costs. More precisely, given an allocation of resources to the agents, let us use $X^\ell \subseteq [n]$ to denote the set of agents that hold resource $\ell$ in their caches. Then, the cost of agent $j$ to get access to resource $\ell$ is given by $w_j^\ell d(j, X^\ell)$, where

$$d(j, X^\ell) = \min\{c_{ij} : i \in X^\ell\},$$

is the minimum distance that agent $j$ needs to travel to get access to resource $\ell$. In particular, the overall access cost among all the agents and for all the resources is given by $\sum_{j,\ell} w_j^\ell d(j, X^\ell)$. An integer program (IP) formulation for the data placement problem is given by

$$\min \sum_{i,j,\ell} w_j^\ell c_{ij} x_{ij}^\ell + \sum_{i,\ell} f_i^\ell y_i^\ell$$

$$x_{ij}^\ell \le y_i^\ell \;\; \forall i, j, \ell,$$

$$\sum_{i=1}^{n} x_{ij}^\ell \ge 1 \;\; \forall j, \ell,$$

$$\sum_{\ell=1}^{k} y_i^\ell \le u_i \;\; \forall i,$$

$$x_{ij}^\ell, y_i^\ell \in \{0, 1\}, \;\; \forall i, j, \ell, \tag{1}$$

where $y_i^\ell = 1$ if we allocate resource $\ell$ to agent $i$, and $x_{ij}^\ell = 1$ if agent $j$ gets access to resource $\ell$ through agent $i$. The first set of constraints in (1) ensures that agent $j$ can access resource $\ell$ through agent $i$ only if agent $i$ holds resource $\ell$ in its cache. The second set of constraints implies that each agent $j$ has to get access to all the resources, and the last set of constraints is the cache capacity constraints that allow agent $i$ to hold at most $u_i$ resources in its cache. Subject to these constraints, the goal is to allocate the resources to the agents to minimize the sum of the placement cost $\sum_{i,\ell} f_i^\ell y_i^\ell$ and the access cost $\sum_{i,j,\ell} w_j^\ell c_{ij} x_{ij}^\ell$.

The following lemma shows that without loss of generality, we can restrict our attention to the data placement problem with unit cache size, i.e., when $u_i = 1 \forall i \in [n]$.

LEMMA 1.1. *An instance of the data placement problem in which agent $i$ has cache size $u_i$, demand vector $(w_i^1, \ldots, w_i^k)$, and installment vector $(f_i^1, \ldots, f_i^k)$ can be reduced to a unit cache size instance by replacing agent $i$ with $u_i$ independent and collocated agents $i_1, \ldots, i_{u_i}$, where each agent $i_j, j = 1, \ldots, u_i$ has a unit cache size, demand vector $(\frac{w_i^1}{u_i}, \ldots, \frac{w_i^k}{u_i})$ and installment vector $(f_i^1, \ldots, f_i^k)$.*

PROOF. Consider an arbitrary allocation profile $X = (X^\ell, \ell \in [k])$ in the original instance and assume that the content of cache $i$ is filled with resources $\ell_1, \ldots, \ell_{u_i}$. Now let us replace agent $i$ with $u_i$ independent and collocated unit cache size agents, where the caches of agents $i_1, \ldots, i_{u_i}$ are filled with $\ell_1, \ldots, \ell_{u_i}$, respectively. For any agent $j \ne i$, the cost of getting $j$ access to all the resources is the same in both instances. Moreover, the cost of agent $i$ in the original instance equals $\sum_\ell w_i^\ell d(i, X^\ell) + \sum_{r=1}^{u_i} f_i^{\ell_r}$, while the total cost of agents $i_1, \ldots, i_{u_i}$ in the new instance equals

$$\sum_\ell \sum_{r=1}^{u_i} w_{i_r}^\ell d(i_r, X^\ell) + \sum_{r=1}^{u_i} f_i^{\ell_r} = \sum_\ell \sum_{r=1}^{u_i} \frac{1}{u_i} w_i^\ell d(i, X^\ell) + \sum_{r=1}^{u_i} f_i^{\ell_r}$$

$$= \sum_\ell w_i^\ell d(i, X^\ell) + \sum_{r=1}^{u_i} f_i^{\ell_r},$$

which shows that the total costs of the two instances are the same. Thus, there is a one-to-one correspondence between optimal solutions of the original instance and the new instance with more unit

cache agents. The proof is completed by repeating the same process for every agent $i$ until all the agents have unit cache size. □

Henceforth, we will focus only on the data placement problem with unit-cache size.[2] But before we get into the analysis, the following proposition shows that even approximating the non-metric data placement problem to within a logarithmic factor is a hard problem (see Appendix I for a proof).

PROPOSITION 1.2. *It is NP-hard to approximate the non-metric data placement problem up to a factor better than $O(\ln n)$.*

REMARK 1. *It is known that when the access costs are metric, i.e., $c_{ik} \le c_{ij} + c_{jk} \; \forall i, j, k \in [n]$, the data placement problem admits constant factor approximations [5, 27]. This is in sharp contrast with the non-metric case that assumes no conditions on the access costs. The reason is that the reduction of Proposition 1.2 is based on non-metric UFLP, while the metric UFLP admits constant factor approximations.*

Despite the above negative result, we are still interested in finding distributed algorithms that perform well in most instances of the non-metric data placement problem. The reason is that often assuming any property on the access costs in general content delivery systems is unrealistic because the access costs could be a complicated function of many factors. To that end, we will develop two distributed algorithms in which either the agents or the resources are viewed as selfish entities that aim to maximize their payoffs, and we will analyze the performance of the allocation profiles resulting from agents' interactions. That not only allows the content delivery system to operate in a fully distributed manner but also allows us to trade-off the computation time to obtain better suboptimal solutions.

## 2 A GAME-THEORETIC DECOMPOSITION FOR THE DATA PLACEMENT PROBLEM

In order to develop a distributed algorithm for the data placement problem, in this section, we first provide a game-theoretic decomposition for its objective function. Let us consider the objective function of the (unit cache size) data placement problem

$$\Phi(x) = \sum_{j,\ell} w_j^\ell d(j, X^\ell) + \sum_i f_i^{x_i},$$

where $x = (x_1, \ldots, x_n) \in [k]^n$ denotes the resource allocation profile of all the agents, $X^\ell = \{i : x_i = \ell\}$ is the set of agents that have resource $\ell$ in their cache, and $d(j, X^\ell) = \min\{c_{ji} : i \in X^\ell\}$. Consider a noncooperative game in which each agent $i \in [n]$ can be viewed as one player with the action set $[k]$. The action of player $i$ is the resource $x_i \in [k]$ that it caches, and incurs a cost that is given by

$$c_i(x) = \sum_{j,\ell} w_j^\ell \big(d(j, X^\ell) - c_{ij}\big)^+ + f_i^{x_i},$$

where for a real number $a$ we define $(a)^+ = \max\{0, a\}$.

LEMMA 2.1. *The above noncooperative game $\mathcal{G} = ([n], [k]^n, \{c_i\})$ is an exact potential game with the potential function $\Phi(x)$.*

---

[2]In fact, the reduction of Lemma 1.1 holds even for metric access costs, i.e., when $c_{ik} \le c_{ij} + c_{jk}, \forall i, j, k$. The reason is that the distances in the unit cache size instance still satisfy the metric property, as the distance between any two agents is the same as the distance between their collocated copies.

Proof. Consider an arbitrary player $i$ and an action (allocation) profile $x = (x_i, x_{-i})$ such that $x_i = \ell$. Assume that player $i$ changes its action from $x_i = \ell$ to $x_i' = \ell'$, and call the new action profile $x' = (x_i', x_{-i})$. For any $o \in [k]$, let us use $X^o$ and $X'^o$ to denote the set of players holding resource $o$ in action profiles $x$ and $x'$, respectively. Then, we have $i \in X^\ell, i \notin X^{\ell'}$ and $X'^\ell = X^\ell - i, X'^{\ell'} = X^{\ell'} + i$. We can write

$$c_i(x_i', x_{-i}) - c_i(x_i, x_{-i})$$
$$= \sum_j w_j^\ell \left(d(j, X'^\ell) - c_{ij}\right)^+ + \sum_j w_j^{\ell'} \left(d(j, X'^{\ell'}) - c_{ij}\right)^+ + f_i^{\ell'}$$
$$- \sum_j w_j^\ell \left(d(j, X^\ell) - c_{ij}\right)^+ - \sum_j w_j^{\ell'} \left(d(j, X^{\ell'}) - c_{ij}\right)^+ - f_i^\ell$$
$$= \sum_j w_j^\ell \left(d(j, X^\ell - i) - c_{ij}\right)^+ + f_i^{\ell'}$$
$$- \sum_j w_j^{\ell'} \left(d(j, X^{\ell'}) - c_{ij}\right)^+ - f_i^\ell, \tag{2}$$

where the first equality holds because for any resource $o \notin \{\ell, \ell'\}$, we have $X^o = X'^o$. The second equality follows from $d(j, X^\ell) \le c_{ij}$ and $d(j, X'^{\ell'}) \le c_{ij}$ because $i \in X^\ell, i \in X'^{\ell'}$.

Next, we compute the amount of change in the potential function $\Phi(x)$. We have

$$\Phi(x_i', x_{-i}) - \Phi(x_i, x_{-i})$$
$$= \sum_j w_j^\ell d(j, X'^\ell) + \sum_j w_j^{\ell'} d(j, X'^{\ell'}) + f_i^{\ell'}$$
$$- \sum_j w_j^\ell d(j, X^\ell) - \sum_j w_j^{\ell'} d(j, X^{\ell'}) - f_i^\ell$$
$$= \sum_j w_j^\ell \left(d(j, X'^\ell) - d(j, X^\ell)\right) + f_i^{\ell'}$$
$$- \sum_j w_j^{\ell'} \left(d(j, X^{\ell'}) - d(j, X'^{\ell'})\right) - f_i^\ell$$
$$= \sum_j w_j^\ell \left(d(j, X^\ell - i) - d(j, X^\ell)\right) + f_i^{\ell'}$$
$$- \sum_j w_j^{\ell'} \left(d(j, X^{\ell'}) - d(j, X^{\ell'} + i)\right) - f_i^\ell$$
$$= \sum_j w_j^\ell \left(d(j, X^\ell - i) - d(j, X^\ell)\right) + f_i^{\ell'}$$
$$- \sum_j w_j^{\ell'} \left(d(j, X^{\ell'}) - c_{ij}\right)^+ - f_i^\ell$$
$$= \sum_j w_j^\ell \left(d(j, X^\ell - i) - c_{ij}\right)^+ + f_i^{\ell'}$$
$$- \sum_j w_j^{\ell'} \left(d(j, X^{\ell'}) - c_{ij}\right)^+ - f_i^\ell, \tag{3}$$

where in the fourth equality we have used the fact that

$$d(j, X^{\ell'}) - d(j, X^{\ell'} + i) = \left(d(j, X^{\ell'}) - c_{ij}\right)^+,$$

by considering two cases. First, if $d(j, X^{\ell'}) \le c_{ij}$, then $d(j, X^{\ell'} + i) = d(j, X^{\ell'})$, and thus $d(j, X^{\ell'}) - d(j, X^{\ell'} + i) = 0 = \left(d(j, X^{\ell'}) - c_{ij}\right)^+$. Second, if $d(j, X^{\ell'}) > c_{ij}$, then $d(j, X^{\ell'} + i) = c_{ij}$, and thus $d(j, X^{\ell'}) - d(j, X^{\ell'} + i) = d(j, X^{\ell'}) - c_{ij} = \left(d(j, X^{\ell'}) - c_{ij}\right)^+$. Similarly,

the last equality is obtained from

$$d(j, X^\ell - i) - d(j, X^\ell) = \left(d(j, X^\ell - i) - c_{ij}\right)^+,$$

which can be shown by considering two cases: if $d(j, X^\ell - i) \le c_{ij}$, then $d(j, X^\ell - i) = d(j, X^\ell)$, and thus $d(j, X^\ell - i) - d(j, X^\ell) = 0 = \left(d(j, X^\ell - i) - c_{ij}\right)^+$. Otherwise, if $d(j, X^\ell - i) > c_{ij}$, then $d(j, X^\ell) = c_{ij}$, and thus $d(j, X^\ell - i) - d(j, X^\ell) = d(j, X^\ell - i) - c_{ij} = \left(d(j, X^\ell - i) - c_{ij}\right)^+$. Finally, by comparing (2) and (3), we get

$$\Phi(x_i', x_{-i}) - \Phi(x_i, x_{-i}) = c_i(x_i', x_{-i}) - c_i(x_i, x_{-i}),$$

which completes the proof. □

As a result of Lemma 2.1, if players selfishly update their resources by minimizing their cost functions, the overall allocation profile will converge to a pure Nash equilibrium (NE), which must be a local minimum of the potential function. Therefore, one could ask about the quality of the solution obtained at a NE compared to the global optimum of the potential function, which is the optimal solution to the data placement problem. To evaluate the quality of a solution obtained at a NE, we leverage the dual program corresponding to the linear program relaxation of the data placement problem (1), which is given by

$$\max \sum_{j,\ell} \beta_j^\ell - \sum_i \alpha_i$$
$$\beta_j^\ell - u_{ij}^\ell \le w_j^\ell c_{ij} \ \forall i, j, \ell,$$
$$\sum_j u_{ij}^\ell - \alpha_i \le f_i^\ell \ \forall i, \ell,$$
$$u_{ij}^\ell, \beta_j^\ell, \alpha_i \ge 0, \ \forall i, j, \ell. \tag{4}$$

Using the first set of constraints, in an optimal dual solution we may assume $u_{ij}^\ell = \left(\beta_j^\ell - w_j^\ell c_{ij}\right)^+, \forall i, j, \ell$. Otherwise, if $u_{ij}^\ell > \left(\beta_j^\ell - w_j^\ell c_{ij}\right)^+$ for some $i, j, \ell$, we can create a new feasible dual solution by reducing $u_{ij}^\ell$ to $\left(\beta_j^\ell - w_j^\ell c_{ij}\right)^+$. Such a change preserves the dual feasibility of the second set of constraints while potentially allowing one to reduce $\alpha_i$ and hence increase the dual objective value. By abuse of notation, if we use $\beta_j^\ell$ to denote $\beta_j^\ell / w_j^\ell$, we can write the dual program (4) in an equivalent form as

$$\max \sum_{j,\ell} w_j^\ell \beta_j^\ell - \sum_i \alpha_i$$
$$\sum_j w_j^\ell \left(\beta_j^\ell - c_{ij}\right)^+ - f_i^\ell \le \alpha_i \ \forall i, \ell,$$
$$\beta_j^\ell, \alpha_i \ge 0, \ \forall i, j, \ell. \tag{5}$$

Using this dual formulation, we are able to show the following bound on the quality of an NE compared to the optimal global solution. The proof of this theorem can be found in Appendix A.

Theorem 2.2. *Let $x$ be any pure NE of the potential game $\mathcal{G}$ and $x^o$ be the optimal solution to the data placement problem. Then,*

$$\Phi(x) \le \frac{\Phi(x^o) + \sum_{j=1}^n \Phi(x \setminus j)}{n + 1},$$

*where $\Phi(x \setminus j)$ denotes the value of the potential function when the cache content of player $j$ is evacuated.*

Theorem 2.2 provides an upper bound for the objective value at any NE in terms of the global minimum value and the objective function's sensitivity to each player's cache content at that NE. For instance, if the resources are well-distributed such that each agent $j$ can find a resource of similar type at a distance of at most $d$, then $\Phi(x \setminus j) \le \Phi(x) + d$, in which Theorem 2.2 implies $\Phi(x) \le \Phi(x^o) + nd$. Note that the optimal solution $x^o$ is also a NE as it is the global maximizer of the potential function. However, due to the non-metric property of the access costs, it may be possible that the potential function $\Phi(\cdot)$ has many local minima (NE points). In that case, in the worst-case scenario, the quality of an arbitrary NE can be significantly smaller than that of the best NE $x^o$. The reason is that a NE is the outcome of a local search algorithm that is unimprovable up to a single-player deviation.[3] Therefore, to bridge this gap at a more computational cost, in the next section, we rely on Glauber dynamics with noisy updates to steer the resource allocation outcome resulting from players' interactions closer to the global optimal solution.

# 3 GLAUBER DYNAMICS FOR FINDING A GLOBAL OPTIMAL SOLUTION

Let $\mathcal{X} = [k]^n$ be the space of all possible resource allocations. We consider Glauber dynamics over the space $\mathcal{X}$ in which players iteratively update their cache contents. More precisely, given an allocation profile $x \in \mathcal{X}$, at each time instance $t = 1, 2, \ldots$, one player $i$ will be chosen uniformly and independently from the past and will update its resource to $o \in [k]$ with probability

$$\frac{e^{-\beta c_i(o, x_{-i})}}{\sum_{\ell \in [k]} e^{-\beta c_i(\ell, x_{-i})}}, \tag{6}$$

where $\beta \in [0, \infty)$ is a noise parameter. In other words, given that player $i$ is chosen to update its resource at time $t$, the probability that it caches resource $o$ is proportional to the utility that resource $o$ brings to that player subject to an additional noise $\beta$ that captures the uncertainty or mistake of player $i$ in choosing resource $o$. As $\beta \to \infty$, the above Glauber dynamics replicate the best response dynamics. Moreover, one can see that the above Glauber dynamics induce a Markov chain over the state space of all the allocation profiles $\mathcal{X}$. The following lemma shows that the stationary distribution of such a Markov chain is given by the Gibbs distribution with respect to the potential function $\Phi$ (see Appendix I for a proof).

LEMMA 3.1. *The stationary distribution of the Markov chain induced by the Glauber dynamics (6) is given by $\pi : \mathcal{X} \to [0, 1]$, where*

$$\pi(x) = \frac{e^{-\beta \Phi(x)}}{\sum_{z \in \mathcal{X}} e^{-\beta \Phi(z)}}. \tag{7}$$

As a result, for sufficiently large $\beta$, the Glauber dynamics will concentrate on an allocation profile with the smallest potential function, which is the global minimum of the data placement problem. However, for larger $\beta$, the induced chain takes longer to converge to its stationary distribution. Nevertheless, the following theorem shows that still for reasonably large values of $\beta$, the induced Markov chain mixes quickly to its stationary Gibbs distribution.

---

[3]In fact, it is known that for the simpler UFLP or $k$-median problem, a richer class of local search moves are required to guarantee the existence of a "good" suboptimal solution [29].

THEOREM 3.2. *Given $\epsilon > 0$, let $t_{\mathrm{mix}}(\epsilon)$ be the $\epsilon$-mixing time of the Glauber dynamics with underlying transition matrix $P : \mathcal{X}^2 \to [0, 1]$ and stationary distribution $\pi$, i.e., $t_{\mathrm{mix}} = \min_t \sup_\mu \|\mu P^t - \pi\|_{TV}$. Then for $\beta \le \frac{k}{6nu}$ where $u = \max_{i,x} c_i(x)$, the mixing time of the Glauber dynamics is at most $t_{\mathrm{mix}}(\epsilon) = O(n \ln \frac{n}{\epsilon})$.*

PROOF. Let $x$ and $y$ be two allocation profiles that differ in the resource of exactly one player $i$, that is, $x_{-i} = y_{-i}$ and $x_i \ne y_i$. Let $Z_t^x$ and $Z_t^y$ be the Markov chains obtained from Glauber dynamics with initial states $x$ and $y$, respectively. Moreover, by abuse of notation, let us use $x$ and $y$ to denote the current states of the two Markov chains, respectively. Assuming that player $i' \in [n]$ is selected to update its action at the current time, the transition probability distributions of the chains denoted by $\mu^{i'}$ and $\nu^{i'}$ equal

$$\mu_o^{i'} = \frac{e^{-\beta c_{i'}(o, x_{-i'})}}{\sum_{\ell \in [k]} e^{-\beta c_{i'}(\ell, x_{-i'})}}, \ o \in [k],$$

$$\nu_o^{i'} = \frac{e^{-\beta c_{i'}(o, y_{-i'})}}{\sum_{\ell \in [k]} e^{-\beta c_{i'}(\ell, y_{-i'})}}, \ o \in [k].$$

We couple these chains together by allowing the same player and the same resource (whenever possible) to be used in both chains at each time instance. More precisely, if player $i' = i$ is selected to update, then in both chains, we update the resource of player $i$ to $o$ with the probability given in (6). Otherwise, if player $i' \ne i$ is selected, we update the resource of player $i'$ in both chains according to the optimal coupling between $\mu^{i'}$ and $\nu^{i'}$.[4]

For two action profiles $z, z' \in \mathcal{X}$, let $\rho(z, z')$ denote the number of positions in which $z$ and $z'$ differ from each other. According to the above coupling, when player $i$ is selected, the two chains become identical, i.e., $\rho(Z_1^x, Z_1^y) = 0$. Thus, $\rho(Z_1^x, Z_1^y)$ might increase from 1 to 2 only if a player $i' \ne i$ were selected, and the resource of that player were updated to two different resources in those chains. Let $M^{i'}$ and $N^{i'}$ be the random variables denoting the updated resource of player $i'$ with corresponding distributions $\mu^{i'}$ and $\nu^{i'}$, respectively. We have

$$\mathbb{P}\{\rho(Z_1^x, Z_1^y) = 2\} = \frac{1}{n} \sum_{i' \ne i} \mathbb{P}\{M^{i'} \ne N^{i'}\} = \frac{1}{n} \sum_{i' \ne i} \|\mu^{i'} - \nu^{i'}\|_{TV},$$

where the second equality holds because we use optimal coupling of distributions $\mu^{i'}$ and $\nu^{i'}$ to update the resource of player $i'$.

Next, we proceed to bound $\|\mu^{i'} - \nu^{i'}\|_{TV}$. Given action profiles $x = (x_i, x_{-i})$ and $y = (y_i, x_{-i})$, by abuse of notation, let $X^{x_i}$ and $X^{y_i}$ be the set of players in $[n] \setminus \{i'\}$ that hold resources $x_i$ and $y_i$ in the action profile $x$, respectively. Then, if $x_{i'} \notin \{x_i, y_i\}$, we have

$$c_{i'}(x) - c_{i'}(y) = \sum_j w_j^{y_i}\big(d(j, X^{y_i}) - c_{i'j}\big)^+ + \sum_j w_j^{x_i}\big(d(j, X^{x_i}) - c_{i'j}\big)^+$$

$$- \sum_j w_j^{y_i}\big(d(j, X^{y_i} + i) - c_{i'j}\big)^+ - \sum_j w_j^{x_i}\big(d(j, X^{x_i} - i) - c_{i'j}\big)^+ := \Delta.$$

Otherwise, if $x_{i'} = x_i$, then

$$c_{i'}(x) - c_{i'}(y)$$

$$= \sum_j w_j^{y_i}\Big(\big(d(j, X^{y_i}) - c_{i'j}\big)^+ - \big(d(j, X^{y_i} + i) - c_{i'j}\big)^+\Big) = \Delta + \Delta_{x_i},$$

---

[4]Given two random variables $X$ and $Y$ with distributions $\pi_X$ and $\pi_Y$, the optimal coupling between them induces a joint probability distribution $\mathbb{P}$ over $(X, Y)$ such that $P(X \ne Y) = \|\pi_X - \pi_Y\|_{TV}$, where $\|\cdot\|_{TV}$ denotes the total variation distance.

where $\Delta_{x_i} := \sum_j w_j^{x_i} \left( \left( d(j, X^{x_i} - i) - c_{i'j} \right)^+ - \left( d(j, X^{x_i}) - c_{i'j} \right)^+ \right) \geq 0$.

Similarly, if $x_{i'} = y_i$, we have

$$c_{i'}(x) - c_{i'}(y)$$
$$= \sum_j w_j^{x_i} \left( \left( d(j, X^{x_i}) - c_{i'j} \right)^+ - \left( d(j, X^{x_i} - i) - c_{i'j} \right)^+ \right) = \Delta - \Delta_{y_i},$$

where $\Delta_{y_i} := \sum_j w_j^{y_i} \left( \left( d(j, X^{y_i}) - c_{i'j} \right)^+ - \left( d(j, X^{y_i} + i) - c_{i'j} \right)^+ \right) \geq 0$.

Now let us define the notations

$$B_1 = e^{-\beta c_{i'}(x_i, x_{-i'})}, \quad B_2 = e^{-\beta c_{i'}(y_i, x_{-i'})}, \quad B = B_1 + B_2,$$
$$C_1 = e^{-\beta(c_{i'}(x_i, x_{-i'}) - \Delta_{x_i})}, \quad C_2 = e^{-\beta(c_{i'}(y_i, x_{-i'}) + \Delta_{y_i})}, \quad C = C_1 + C_2,$$
$$A = \sum_{\ell \notin \{x_i, y_i\}} e^{-\beta c_{i'}(\ell, x_{-i'})},$$

and note that $B_2 \geq C_2$ and $C_1 \geq B_1$. Then, the probability distribution $\nu^{i'}$ can be written as

$$\nu_o^{i'} = \begin{cases} \frac{\exp(-\beta c_{i'}(o, x_{-i'}))}{A + C} & \text{if } o \notin \{x_i, y_i\}, \\ \frac{C_1}{A + C} & \text{if } o = x_i, \\ \frac{C_2}{A + C} & \text{if } o = y_i. \end{cases}$$

Therefore, by definition of the total variation, we have

$$2\|\nu^{i'} - \mu^{i'}\|_{TV} = |\nu_{x_i}^{i'} - \mu_{x_i}^{i'}| + |\nu_{y_i}^{i'} - \mu_{y_i}^{i'}| + \sum_{o \notin \{x_i, y_i\}} |\nu_o^{i'} - \mu_o^{i'}|. \tag{8}$$

To bound the last term in (8), for any $o \notin \{x_i, y_i\}$, we have

$$\sum_{o \notin \{x_i, y_i\}} |\nu_o^{i'} - \mu_o^{i'}| = \sum_{o \notin \{x_i, y_i\}} \left| \frac{e^{-\beta c_{i'}(o, x_{-i'})}}{A + C} - \frac{e^{-\beta c_{i'}(o, x_{-i'})}}{A + B} \right|$$
$$= \frac{A|B - C|}{(A + C)(A + B)}.$$

Similarly, we can compute the first two terms in (8) as

$$|\nu_{x_i}^{i'} - \mu_{x_i}^{i'}| = \frac{C_1}{A + C} - \frac{B_1}{A + B},$$
$$|\nu_{y_i}^{i'} - \mu_{y_i}^{i'}| = \frac{B_2}{A + B} - \frac{C_2}{A + C}.$$

Substituting the above three relations into (8), we get

$$2\|\nu^{i'} - \mu^{i'}\|_{TV} = \frac{B_2 - B_1}{A + B} + \frac{C_1 - C_2}{A + C} + \frac{A|B - C|}{(A + C)(A + B)}.$$

Let us define $u = \max_{i,x} c_i(x)$ and note that $\Delta_{x_i} \leq u$ and $\Delta_{y_i} \leq u$. Then, $A + B \geq ke^{-\beta u}$ and $A + C \geq ke^{-2\beta u}$. Using the mean-value theorem for $f(r) = e^{-\beta r}$, we have the following relations:

$$B_2 - B_1 \leq \beta |c_{i'}(x_i, x_{-i'}) - c_{i'}(y_i, x_{-i'})| e^0 \leq \beta u,$$
$$C_1 - C_2 \leq \beta |c_{i'}(x_i, x_{-i'}) - \Delta_{x_i} - c_{i'}(y_i, x_{-i'}) - \Delta_{y_i}| e^{\beta u} \leq 3\beta u e^{\beta u},$$
$$|B - C| \leq |B_1 - C_1| + |B_2 - C_2| \leq 2\beta u e^{\beta u}.$$

Therefore, for any $i'$ we have

$$2\|\nu^{i'} - \mu^{i'}\|_{TV} \leq \frac{\beta u}{k} e^{\beta u} + \frac{3\beta u}{k} e^{3\beta u} + \frac{2\beta u}{k} e^{2\beta u} \leq \frac{6\beta u}{k} e^{3\beta u}$$
$$\Rightarrow \|\nu^{i'} - \mu^{i'}\|_{TV} \leq \frac{3\beta u}{k} e^{3\beta u}. \tag{9}$$

Next, we bound the mixing time of the Glauber dynamics. For one step of the chain, we have

$$\mathbb{E}[\rho(Z_1^x, Z_1^y)] = 1 - \frac{1}{n} + \frac{1}{n} \sum_{i' \neq i} \|\mu^{i'} - \nu^{i'}\|_{TV}. \tag{10}$$

Substituting (9) into (10) and using the assumption $\beta \leq \frac{k}{6nu}$, we get

$$\mathbb{E}[\rho(Z_1^x, Z_1^y)] \leq 1 - \frac{1}{n} + \frac{n-1}{nk} 3\beta u e^{3\beta u}$$
$$\leq 1 - \frac{1}{n} + \frac{1}{k}(3\beta u) e^{3\beta u}$$
$$\leq 1 - \frac{1}{n} + \frac{1}{2n} e^{\frac{k}{4n}} \leq 1 - \frac{1}{7n}.$$

Starting from any two arbitrary initial states $x$ and $y$ that differ in $d$ positions, we can reach from $x$ to $y$ using a sequence $x^0 = x, x^1, \ldots, x^d = y$ such that every two consecutive allocations differ in exactly one position. As $\rho(\cdot, \cdot)$ is metric over the space of allocations, using triangle inequality, we can write

$$\mathbb{E}[\rho(Z_1^x, Z_1^y)] \leq \sum_{k=1}^d \mathbb{E}[\rho(Z_1^{x^{k-1}}, Z_1^{x^k})]$$
$$\leq (1 - \frac{1}{7n})d = (1 - \frac{1}{7n})\rho(x, y).$$

Moreover, using the Markov property of the chains, we can write

$$\mathbb{E}[\rho(Z_t^x, Z_t^y)] = \mathbb{E}\left[\mathbb{E}[\rho(Z_t^x, Z_t^y)|Z_{t-1}^x, Z_{t-1}^y]\right]$$
$$= \mathbb{E}[\rho(Z_1^{Z_{t-1}^x}, Z_1^{Z_{t-1}^y})]$$
$$\leq (1 - \frac{1}{7n})\mathbb{E}[\rho(Z_{t-1}^x, Z_{t-1}^y)].$$

By using the above inequality recursively, we obtain

$$\mathbb{E}[\rho(Z_t^x, Z_t^y)] \leq (1 - \frac{1}{7n})^t \rho(Z_0^x, Z_0^y) = (1 - \frac{1}{7n})^t \rho(x, y) \leq ne^{-\frac{t}{7n}}.$$

Finally, using Markov's inequality, we can write

$$\mathbb{P}(Z_t^x \neq Z_t^y) \leq \mathbb{P}(\rho(Z_t^x, Z_t^y) \geq 1) \leq \mathbb{E}[\rho(Z_t^x, Z_t^y)] \leq ne^{-\frac{t}{7n}}.$$

The above relation, in view of Lemma .4, shows that the mixing time of the Gluaber dynamics is at most $t_{\mathrm{mix}}(\epsilon) = O(n \ln \frac{n}{\epsilon})$. □

As we mentioned earlier, there is a trade-off between the mixing time of the Glauber dynamics and the concentration of the induced stationary distribution around the global optimum solution. For the Glauber dynamics to concentrate better around the optimal solution of the data placement problem, one needs a higher noise parameter $\beta$. However, choosing $\beta$ too large can result in a slow mixing time. Therefore, Theorem 3.2 provides a threshold for this computation/optimality trade-off by characterizing a range of noise parameters with a fast mixing time guarantee, while for obtaining better suboptimal solutions, one must pay the price with higher computation time. Note that such a computation/optimality trade-off is inevitable because by Proposition 1.2, finding a suboptimal solution within a factor better than $O(\log n)$ will likely require supper-polynomial running time. However, in practice, the costs of players are mainly determined by their nearby neighbors. Thus, one can leverage the locality of players' cost functions to establish a fast mixing time for larger values of $\beta$. For instance, if each player's action can affect the cost of at most $d$ nearby players, then the bound for $\beta$ in Theorem 3.2 can be improved to $\beta = O(\frac{1}{u} \ln(\frac{k}{d}))$.

# 4 AUCTION-BASED DISTRIBUTED METHOD FOR THE DATA PLACEMENT PROBLEM

In the previous section, we developed a distributed game-theoretic framework that allows players to update their resources selfishly subject to some noise parameter. In that formulation, players are the agents, and the actions are the choices of resources. An alternative perspective is to view the resources as players who bid to buy the cache spaces of the agents (viewed as items). That leads us to the following auction-based distributed algorithm for the data placement problem.

## 4.1 An Auction-Based Distributed Algorithm

Consider an auction with $k$ players (resources) and $n$ items. We view the unit cache space of agent $i$ as an item that will be sold to players. We assume that each player $\ell \in [k]$ represents a set of $n$ clients $\{(j, \ell) : j \in [n]\}$ and charges a $\beta_j^\ell \geq 0$ fee per unit demand to client $(j, \ell)$. This charge is for representing client $(j, \ell)$ in the auction and for connecting that client to resource $\ell$. Moreover, we assume that the items are sold separately using a first-price auction in which players submit their bids for different items. An item is sold to the player with the highest bid (ties are broken arbitrarily), and the winner must pay an amount equal to the highest bid. In addition, we assume that the entrance fee for player $\ell$ to participate in the auction for item $i$ is $f_i^\ell$. Next, we specify the bidding strategies for the players.

Let us consider player $\ell$, who charges $\beta_j^\ell$ per unit demand to its client $(j, \ell)$. From that amount, player $\ell$ subtracts $c_{ij}$ to account for the cost of connecting $(j, \ell)$ to agent $i$, and therefore includes only a $(\beta_j^\ell - c_{ij})^+$ portion of that amount toward bidding for item $i$. Therefore, summing over the total demand of all the clients, player $\ell$ bids $\left( \sum_j w_j^\ell (\beta_j^\ell - c_{ij})^+ - f_i^\ell \right)^+$ toward item $i$, where the term $f_i$ is to account for the entrance fee that player $\ell$ has to pay to be able to bid for item $i$. Thus, if player $\ell$ wins a bundle of items $X^\ell \subseteq [n]$ in the auction, $\ell$'s utility equals the amount that $\ell$ collects from its clients minus the amount that $\ell$ has to pay to the auctioneer, i.e.,

$$u_\ell(\beta^\ell, X^\ell) = \sum_j w_j^\ell \beta_j^\ell - \sum_{i \in X^\ell} \left( \sum_j w_j^\ell (\beta_j^\ell - c_{ij})^+ - f_i^\ell \right)^+$$
$$= \sum_j w_j^\ell \beta_j^\ell - \sum_{i \in X^\ell} \left( \sum_j w_j^\ell (\beta_j^\ell - c_{ij})^+ - f_i^\ell \right), \quad (11)$$

where the second equality holds by individual rationality. Otherwise, if $\sum_j w_j^\ell (\beta_j^\ell - c_{ij})^+ - f_i^\ell < 0$ for some $i$, there is no incentive for player $\ell$ to enter the auction for item $i$. Therefore, player $\ell$'s goal is to determine a charging strategy $\beta_j^\ell$ to maximize its utility.

## 4.2 Performance Guarantee of the Solution

To analyze the performance guarantee of the allocation profile obtained from the above auction, let us again consider the dual program corresponding to the LP relaxation of (1) given by

$$\max \sum_{j,\ell} w_j^\ell \beta_j^\ell - \sum_i \alpha_i$$
$$\sum_j w_j^\ell (\beta_j^\ell - c_{ij})^+ - f_i^\ell \leq \alpha_i \ \forall i, \ell,$$
$$\beta_j^\ell, \alpha_i \geq 0, \ \forall i, j, \ell. \quad (12)$$

To satisfy all the constraints in (12) while maximizing the dual objective function, we must set $\alpha_i = \max_\ell \left( \sum_j w_j^\ell (\beta_j^\ell - c_{ij})^+ - f_i^\ell \right)^+$, which gives us the following compact form for the dual program:

$$\max_{\beta_j^\ell \geq 0} \left\{ \sum_{j,\ell} w_j^\ell \beta_j^\ell - \sum_i \max_\ell \left( \sum_j w_j^\ell (\beta_j^\ell - c_{ij})^+ - f_i^\ell \right)^+ \right\}. \quad (13)$$

Let us use variables $y_i^\ell$ to denote the inner maximization in (13) as[5]

$$\max_{\substack{\beta_j^\ell \geq 0 \ \sum_\ell y^\ell \leq 1 \\ y^\ell \geq 0 \forall \ell}} \min \left\{ \sum_{j,\ell} w_j^\ell \beta_j^\ell - \sum_i \sum_\ell y_i^\ell \left( \sum_j w_j^\ell (\beta_j^\ell - c_{ij})^+ - f_i^\ell \right) \right\}. \quad (14)$$

Now, for every resource type $\ell$, let us define a utility function as

$$u_\ell(\beta^\ell, y^\ell) = \sum_j w_j^\ell \beta_j^\ell - \sum_i y_i^\ell \left( \sum_j w_j^\ell (\beta_j^\ell - c_{ij})^+ - f_i^\ell \right),$$

which is the same as the utility function (11) defined for player $\ell$ if we take $X^\ell = \{i : y_i^\ell = 1\}$. Then, the dual program (14) becomes

$$\max_{\substack{\beta_j^\ell \geq 0 \ \sum_\ell y^\ell \leq 1 \\ y^\ell \geq 0 \forall \ell}} \min \sum_\ell u_\ell(\beta^\ell, y^\ell) = \min_{\substack{\sum_\ell y^\ell \leq 1 \\ y^\ell \geq 0 \forall \ell}} \max_{\beta_j^\ell \geq 0} \sum_\ell u_\ell(\beta^\ell, y^\ell), \quad (15)$$

where the equality holds because each $u_\ell(\beta^\ell, y^\ell)$ is concave in $\beta^\ell$ and linear (convex) in $y^\ell$. In particular, we note that the optimal solution to $y$ is always integral because $u_\ell(\beta^\ell, y^\ell)$ is linear with respect to $y^\ell$ and the constraints $\{\sum_\ell y^\ell \leq 1, y^\ell \geq 0 \forall \ell\}$ define an integral polytope.

Using the above derivations, it should be clear that if $(\beta, y)$ is an optimal dual solution to (15), then player $\ell$'s strategy to maximize its utility is to charge $w_j^\ell \beta_j^\ell$ to client $(j, \ell)$, and $\ell$ receives item $i$ if $y_i^\ell = 1$, in which case $\ell$ has to pay $\alpha_i = \sum_j w_j^\ell (\beta_j^\ell - c_{ij})^+ - f_i^\ell$, which is the maximum bid among all the bids for item $i$. Therefore, the allocation profile obtained from the auction when all the players selfishly maximize their utilities is the same as the optimal dual solution to the min-max problem (15).

THEOREM 4.1. *Consider the data placement problem with zero placement costs $f_i^\ell = 0, \forall i, \ell$, and let $(\alpha, \beta)$ be the optimal solution to the dual program (4). Then, the resource allocation profile obtained from the auction is an $(\frac{1}{1-\gamma})$-approximation of the data placement problem, where $\gamma = \|\alpha\|_1 / \|\beta\|_1 \in [0, 1)$.*

PROOF. We prove the theorem through the following four steps:
**I) Primal feasibility and integrality:** By abuse of notation, let $(\beta, y)$ be the minimal optimal solution to the min-max dual problem (15) (i.e., a solution with the least number of nonzero entries), and let $X^\ell = \{i : y_i^\ell = 1\}$. Clearly, $(X^\ell, \ell \in [k])$ partitions the set of agents $[n]$. We complement this solution with an integral feasible solution for the primal program by setting $x_{ij}^\ell = 1$ if $i = \text{argmin}_{i' \in X^\ell} c_{i'j}$ (ties are broken arbitrarily), i.e., we connect agent $j$ to the closest agent in $X^\ell$ to get access to resource $\ell$. Then, $(x, y)$ defined in this way forms a feasible integral solution to the primal program (1).
**II) Dual feasibility:** Let $\beta$ be the optimal solution to (15) and define $\alpha_i = \max_\ell \left( \sum_j w_j^\ell (\beta_j^\ell - c_{ij})^+ - f_i^\ell \right)^+$ and $u_{ij}^\ell = w_j^\ell (\beta_j^\ell - c_{ij})^+$. Then,

---

[5]In fact, $y_i^\ell$ can be thought of as dual variables corresponding to the first set of constraints in the convex program (12), which also coincide with the primal variables $y_i^\ell$ in the original linear program (1).

from the above arguments, $(\{\alpha_i\}, \{w_j^\ell \beta_j^\ell\}, \{u_{ij}^\ell\})$ forms an optimal solution to the dual program (4).

**III) Complementary slackness conditions:** Based on Proposition 1.2, we cannot expect all the complementary slackness conditions for the above primal-dual solutions to hold.[6] However, as we show, the proposed solutions satisfy most of these conditions and still constitute a good suboptimal solution.

1) Since $y$ is a minimal optimal integral solution to (14), $y_i^\ell = 1$ implies that

$$\alpha_i = \max_{\ell'} \Big( \sum_j w_j^{\ell'} (\beta_j^{\ell'} - c_{ij})^+ - f_i^{\ell'} \Big)^+$$
$$= \Big( \sum_j w_j^\ell (\beta_j^\ell - c_{ij})^+ - f_i^\ell \Big)^+$$
$$= \sum_j w_j^\ell (\beta_j^\ell - c_{ij})^+ - f_i^\ell,$$

where the last equality holds because if $\sum_j w_j^\ell (\beta_j^\ell - c_{ij})^+ - f_i^\ell < 0$ we would have $y_i^\ell = 0$.

2) Since the primal solution $(X^\ell, \ell \in [k])$ partitions $[n]$, and using the definition of $x$ that assigns every agent $j$ to exactly one agent in each $X^\ell$, the primal constraints $\sum_\ell y_i^\ell \leq 1$ and $\sum_i x_{ij}^\ell \geq 1$ are always satisfied by equality. Therefore, the complementary slackness conditions always hold for these two types of primal constraints.

3) As we showed before, for the optimal dual solution we have $u_{ij}^\ell = w_j^\ell (\beta_j^\ell - c_{ij})^+ \forall i, j, \ell$. Therefore, to show complementary slackness for the first set of dual constraints in (4), we only need to show that if $x_{i'j'}^\ell = 1$ for some $i', j', \ell$, then $(\beta_{j'}^\ell - c_{i'j'})^+ = \beta_{j'}^\ell - c_{i'j'}$, or, equivalently, $\beta_{j'}^\ell \geq c_{i'j'}$. This is also true because if $x_{i'j'}^\ell = 1$, that means $y_{i'}^\ell = 1$ (and thus $i' \in X^\ell$) and $c_{i'j'} \leq c_{ij'}, \forall i \in X^\ell$. Moreover, using case (1) we have $\alpha_i = \sum_j w_j^\ell (\beta_j^\ell - c_{ij})^+ - f_i^\ell \geq 0, \forall i \in X^\ell$. Therefore,

$$u_\ell(\beta^\ell, X^\ell) = \sum_j w_j^\ell \beta_j^\ell - \sum_{i \in X^\ell} \Big( \sum_j w_j^\ell (\beta_j^\ell - c_{ij})^+ - f_i^\ell \Big)$$
$$= \sum_{i \in X^\ell} f_i^\ell + \sum_j w_j^\ell \beta_j^\ell - \sum_j \sum_{i \in X^\ell} w_j^\ell (\beta_j^\ell - c_{ij})^+$$
$$= \sum_{i \in X^\ell} f_i^\ell + \sum_{j \neq j'} \Big( w_j^\ell \beta_j^\ell - \sum_{i \in X^\ell} w_j^\ell (\beta_j^\ell - c_{ij})^+ \Big)$$
$$+ \Big( w_{j'}^\ell \beta_{j'}^\ell - \sum_{i \in X^\ell} w_{j'}^\ell (\beta_{j'}^\ell - c_{ij'})^+ \Big).$$

Suppose, by contrary, $\beta_{j'}^\ell < c_{i'j'}$. Then $\beta_{j'}^\ell < c_{ij'} \forall i \in X^\ell$ and we have $\sum_{i \in X^\ell} w_{j'}^\ell (\beta_{j'}^\ell - c_{ij'})^+ = 0$. Therefore, if $\beta_{j'}^\ell$ is slightly increased, the last term in the above expression strictly increases.[7] This contradicts the fact that $\beta^\ell$ corresponds to the optimal dual solution that maximizes $u_\ell(\cdot, X^\ell)$.

**IV) Bounding the performance:** Using the properties of the primal-dual solutions that we established above, the only set of constraints that may violate the complementary slackness conditions are the primal constraints $x_{ij}^\ell \leq y_i^\ell$ with the corresponding dual variables $u_{ij}^\ell = w_j^\ell (\beta_j^\ell - c_{ij})^+$. Therefore, using Lemma .5 with the block of constraints $A_1 \tilde{x} \geq b_1$ representing constraints $y_i^\ell - x_{ij}^\ell \geq 0$ and corresponding dual variables $u_1^* = (u_{ij}^\ell)$, the cost of the generated primal solution denoted by $\mathrm{Cost}(X)$ equals

$$\mathrm{Cost}(X) = \mathrm{OPT} + \sum_{i,j,\ell} u_{ij}^\ell (y_i^\ell - x_{ij}^\ell) \leq \mathrm{OPT} + \sum_{i,j,\ell} u_{ij}^\ell (1 - 0)$$
$$= \mathrm{OPT} + \sum_\ell \sum_{i \in X^\ell} \sum_j w_j^\ell (\beta_j^\ell - c_{ij})^+$$
$$= \mathrm{OPT} + \sum_\ell \sum_{i \in X^\ell} (\alpha_i + f_i^\ell)$$
$$= \mathrm{OPT} + \sum_i \alpha_i, \tag{16}$$

where the third equality holds through use of case 1 of the complementary slackness conditions, and the last equality holds by the assumption $f_i^\ell = 0, \forall i, \ell$. Dividing both sides by $\mathrm{OPT} = \sum_{j,\ell} \beta_j^\ell - \sum_i \alpha_i$,[8] and using the definition of $\gamma$, completes the proof. □

REMARK 2. *We note that the only place that we assumed $f_i^\ell = 0$ is in the last equality of* (16). *Otherwise, all the remaining derivations continue to hold for general $f_i^\ell \geq 0$. In fact, using the same ideas as in Proposition 1.2, one can show that even under the assumption $f_i^\ell = 0$, the non-metric data placement problem remains NP-hard and inapproximable up to an $O(\log n)$ factor.*

REMARK 3. *In fact, the optimal dual objective value denoted by OPT equals the social welfare resulting from the auction, i.e., the sum of the players' utilities $\mathrm{SW} := \sum_\ell u_\ell = \sum_{j,\ell} w_j^\ell \beta_j^\ell - \sum_i \alpha_i$, while the revenue derived by the auctioneer equals to the sum of all the payments $\mathrm{Rev} := \sum_i \alpha_i$. Therefore, another way of interpreting the result of Theorem (4.1) is to say that the approximation guarantee of the allocation profile obtained from the auction is $1 + \frac{\mathrm{Rev}}{\mathrm{SW}}$.*

## 5 CONCLUSIONS

We studied the general non-metric data placement problem from a multiagent game-theoretic perspective and devised distributed computation algorithms for obtaining or approximating its global optimal solutions. The motivation behind this work is that in many real-world content delivery applications, such as web caches, P2P networks, or ad hoc storage systems, the servers are independent or selfish entities that only want to maximize their own payoffs, yet the goal is to achieve good global performance in terms of content delivery and resource availability. We showed that although the problem is hard to approximate within a logarithmic factor, some natural Glauber dynamics by the servers/agents can collectively result in good suboptimal allocation profiles. Moreover, the achieved suboptimal solutions can continuously get closer to the global optimum solution of the data placement problem at a higher computational time. Finally, we provided an auction-based distributed algorithm that can approximate the global optimum solution with a theoretical performance guarantee and can be easily implemented in distributed content delivery systems.

---

[6] Otherwise, the proposed primal solution will be an optimal integral solution to an NP-hard problem.

[7] Note that since the dual function is given by the sum of utilities defined over separate variables $\beta^\ell, \ell \in [k]$, such an increase does not affect other terms in the dual objective.

[8] Here, we are using the original definition of dual variables $\beta_j^\ell$ given in (4) rather that their scaled version $w_j^\ell \beta_j^\ell$.

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

# APPENDIX I

**Proposition .1.** *It is NP-hard to approximate the non-metric data placement problem up to a factor better than $O(\ln n)$.*

**Proof.** We show that the non-metric uncapacitated facility location problem (UFLP) can be formulated as a special instance of the non-metric data placement problem. On the other hand, it is known that the non-metric UFLP with $n$ clients is at least as hard as the set cover problem, which is hard to approximate within an $O(\ln n)$ factor [17]. Therefore, the same inapproximability result must also hold for the non-metric data placement problem.

Consider an arbitrary instance of the non-metric UFLP with the same set $[n]$ of clients and facilities, non-metric access costs $\{c_{ij} : i, j \in [n]\}$, and facility installment costs $\{f_i, i \in [n]\}$. The goal is to open a subset of facilities and connect each client to its nearest open facility to minimize the sum of the cost of opening facilities and the cost of assigning clients to opened facilities. This problem can be formulated as an instance of the data placement problem with a set $[n]$ of agents, non-metric access costs $\{c_{ij} : i, j \in [n]\}$, and $k = 2$ resources. For the first resource we set $w_j^1 = 1, f_i^1 = f_i, \forall i, j \in [n]$. For the second (dummy) resource we set $w_j^2 = 0, f_i^2 = 0, \forall i, j \in [n]$. In other words, the agents that receive resource $\ell = 1$ correspond to the set of open facilities in the UFLP, while the agents that receive the dummy resource $\ell = 2$ correspond to the set of closed facilities. By the construction, it should be clear that any optimal solution to the non-metric UFLP corresponds to an optimal solution in the non-metric data placement problem with the same objective cost, and vice versa. □

**Lemma .2.** *The stationary distribution of the Markov chain induced by the Glauber dynamics is given by $\pi : \mathcal{X} \to [0, 1]$, where*

$$\pi(x) = \frac{e^{-\beta\Phi(x)}}{\sum_{z \in \mathcal{X}} e^{-\beta\Phi(z)}}. \tag{17}$$

**Proof.** We first note that any transition of the Markov chain is between two states that differ in the resource of at most one player. We show that the distribution (17) satisfies the *detailed-balanced* conditions [21], and hence must be a stationary distribution for the induced Markov chain. Let us consider two allocation profiles $x$ and $y$ that differ in the resource of at most one player $i$, that is, $x_{-i} = y_{-i}$. Then, we have

$$\begin{aligned}
\pi(x)P_{xy} &= \pi(x)\frac{\frac{1}{n}e^{-\beta c_i(y_i, x_{-i})}}{\sum_{\ell \in [k]} e^{-\beta c_i(\ell, x_{-i})}} \\
&= \pi(x)\frac{\frac{1}{n}e^{-\beta(c_i(y_i, x_{-i}) - c_i(x))}}{\sum_{\ell \in [k]} e^{-\beta(c_i(\ell, x_{-i}) - c_i(x))}} \\
&= \pi(x)\frac{\frac{1}{n}e^{-\beta(\Phi(y_i, x_{-i}) - \Phi(x))}}{\sum_{\ell \in [k]} e^{-\beta(\Phi(\ell, x_{-i}) - \Phi(x))}} \\
&= \frac{1}{n}\Big(\frac{e^{-\beta\Phi(x)}}{\sum_{z \in \mathcal{X}} e^{-\beta\Phi(z)}}\Big)\Big(\frac{e^{-\beta\Phi(y)}}{\sum_{\ell \in [k]} e^{-\beta\Phi(\ell, x_{-i})}}\Big).
\end{aligned}$$

Similarly, one can show that

$$\pi(y)P_{yx} = \frac{\frac{1}{n}e^{-\beta\Phi(y)}}{\sum_{z\in\mathcal{X}}e^{-\beta\Phi(z)}}\frac{e^{-\beta c_i(x_i, x_{-i})}}{\sum_{\ell\in[k]}e^{-\beta c_i(\ell, x_{-i})}}$$

$$= \frac{1}{n}\Big(\frac{e^{-\beta\Phi(x)}}{\sum_{z\in\mathcal{X}}e^{-\beta\Phi(z)}}\Big)\Big(\frac{e^{-\beta\Phi(y)}}{\sum_{\ell\in[k]}e^{-\beta\Phi(\ell, x_{-i})}}\Big).$$

Comparing the above two relations shows that $\pi(x)P_{xy} = \pi(y)P_{yx}$, which completes the proof. □

THEOREM .3. *Let $x$ be any pure NE of the potential game $\mathcal{G}$ and $x^o$ be the optimal solution to the data placement problem. Then,*

$$\Phi(x) \le \frac{\Phi(x^o) + \sum_j \Phi(x\setminus j)}{n+1},$$

*where $\Phi(x\setminus j)$ denotes the value of the potential function when the cache content of player $j$ is evacuated.*

PROOF. Let us use $x = (x_i, x_{-i})$ to denote a pure NE of the potential game $\mathcal{G}$. Then, for any player $i$ and any action $x_i'$, if we let $x' = (x_i', x_{-i})$, we must have $c_i(x) \le c_i(x')$, which implies

$$\sum_j w_j^{x_i}\big(d(j, X^{x_i}) - c_{ij}\big)^+ + \sum_j w_j^{x_i'}\big(d(j, X^{x_i'}) - c_{ij}\big)^+ + f_i^{x_i}$$

$$\le \sum_j w_j^{x_i}\big(d(j, X^{x_i} - i) - c_{ij}\big)^+ + \sum_j w_j^{x_i'}\big(d(j, X^{x_i'} + i) - c_{ij}\big)^+ + f_i^{x_i'}.$$

Since $i \in X^{x_i}$ and $i \in X^{x_i'} + i$, for any $i$ and $x_i'$, we have

$$\sum_j w_j^{x_i'}\big(d(j, X^{x_i'}) - c_{ij}\big)^+ - f_i^{x_i'} \le \sum_j w_j^{x_i}\big(d(j, X^{x_i} - i) - c_{ij}\big)^+ - f_i^{x_i}.$$

That means that if we define

$$\beta_j^\ell = d(j, X^\ell) \ge 0,$$
$$\alpha_i = \sum_j w_j^{x_i}\big(d(j, X^{x_i} - i) - c_{ij}\big)^+ - f_i^{x_i},$$

then $(\alpha_i, \beta_j^\ell)$ forms a feasible dual solution to the dual program (5) whose objective value by weak duality is less than the optimal fractional solution to the LP relaxation of (1). Therefore, if the optimal solution of the data placement problem is denoted by $x^o$ with minimum objective cost $\Phi(x^o)$, we have

$$\Phi(x^o) \ge \sum_{j,\ell} w_j^\ell \beta_j^\ell - \sum_i \alpha_i$$

$$= \sum_{j,\ell} w_j^\ell d(j, X^\ell) + \sum_i f_i^{x_i}$$

$$- \sum_i \sum_j w_j^{x_i}\big(d(j, X^{x_i} - i) - c_{ij}\big)^+ - \sum_i f_i^{x_i}$$

$$= \Phi(x) - \sum_i \sum_j w_j^{x_i}\big(d(j, X^{x_i} - i) - c_{ij}\big)^+.$$

As a result, the objective value of the solution obtained at NE $x$ is at most

$$\Phi(x) \le \Phi(x^o) + \sum_i \sum_j w_j^{x_i}\big(d(j, X^{x_i} - i) - c_{ij}\big)^+$$

$$= \Phi(x^o) + \sum_\ell \sum_{i\in X^\ell} \sum_j w_j^\ell\big(d(j, X^\ell - i) - c_{ij}\big)^+$$

$$= \Phi(x^o) + \sum_\ell \sum_j w_j^\ell\Big(\sum_{i\in X^\ell}\big(d(j, X^\ell - i) - c_{ij}\big)^+\Big)$$

$$= \Phi(x^o) + \sum_\ell \sum_j w_j^\ell\big(d(j, X^\ell - i_j) - d(j, X^\ell)\big),$$

where $i_j = \arg\min_{k\in X^\ell} c_{jk}$, and the last equality holds because for any $i \in X^\ell - i_j$, we have $d(j, X^\ell - i) = c_{ji_j} = d(j, X^\ell) \le c_{ij}$, and hence $\big(d(j, X^\ell - i) - c_{ij}\big)^+ = 0$. Thus, if we let $\Phi(x\setminus j)$ be the value of the potential function when the cache content of player $j$ is evacuated, from the above expression we have

$$\Phi(x) \le \Phi(x^o) + \sum_\ell \sum_j w_j^\ell\big(d(j, X^\ell - i_j) - d(j, X^\ell)\big)$$

$$= \Phi(x^o) + \sum_j \big(\Phi(x\setminus j) - \Phi(x)\big),$$

or, equivalently, $\Phi(x) \le \frac{\Phi(x^o) + \sum_j \Phi(x\setminus j)}{n+1}$. □

LEMMA .4. *Let $Z_t^x$ and $Z_t^y$ be copies of a Markov chain with initial states $x$ and $y$ and transition probability matrix $P$. Suppose that for each pair of initial states $x, y \in \mathcal{X}$ there is a coupling $(Z_t^x, Z_t^y)$. Then, $d(t) \le \max_{x,y} \mathbb{P}(Z_t^x \ne Z_t^y)$, where $d(t) = \sup_\mu \|\mu P^t - \pi\|_{TV}$ is the maximum total variation between the distribution of the Markov chain at time $t$ and its stationary distribution $\pi$. In particular, the mixing time of the Markov chains is at most*

$$t_{\mathrm{mix}}(\epsilon) := \min\{t : d(t) < \epsilon\} \le \min\{t : \max_{x,y} \mathbb{P}(Z_t^x \ne Z_t^y) < \epsilon\}.$$

PROOF. The proof follows from Theorem 5.4 and Corollary 5.5 in [21]. □

LEMMA .5. *Consider an LP: $OPT = \min\{cx : Ax \ge b, x \ge 0\}$ and its dual $\max\{ub : uA \le c, u \ge 0\}$. Suppose $A = \big[\frac{A_1}{A_2}\big]$ can be represented using two blocks of constraints $A_1$ and $A_2$. Let $u^*$ be the optimal dual solution, and assume $\tilde{x}$ is a feasible primal solution such that $(\tilde{x}, u^*)$ satisfy all the complementary slackness conditions except for the constraints $A_1\tilde{x} \ge b_1$ with corresponding dual variables $u_1^*$, where $b = \big[\frac{b_1}{b_2}\big]$. Then $\tilde{x}$ forms an approximate optimal solution for the LP such that $c\tilde{x} = OPT + u_1^*(A_1\tilde{x} - b_1)$.*

PROOF. Since dual constraints satisfy complementary slackness with respect to $\tilde{x}$, $(u^*A - c)\tilde{x} = 0$. Moreover, since all the primal constraints except $A_1\tilde{x} \ge b_1$ satisfy complementary slackness conditions,

$$u^*(A\tilde{x} - b) = u_1^*(A_1\tilde{x} - b_1) + u_2^*(A_2\tilde{x} - b_2) = u_1^*(A_1\tilde{x} - b_1),$$

$$\Rightarrow u^*A\tilde{x} = u^*b + u_1^*(A_1\tilde{x} - b_1).$$

Thus, we conclude that $\tilde{x}$ is a feasible primal solution whose objective cost equals

$$c\tilde{x} = u^*A\tilde{x} = u^*b + u_1^*(A_1\tilde{x} - b_1) = OPT + u_1^*(A_1\tilde{x} - b_1),$$

where the last equality follows by strong duality. □

