# OpenReview forum: "Distributed Data Placement and Content Delivery in Web Caches with Non-Metric Access Costs"
_ACM.org/TheWebConf/2024/Conference — TheWebConf24_

### Official Review · Reviewer_bhAg · 2023-10-31

**Novelty:** 4
**Technical Quality:** 4

**Review:**

This submission tries to study distributed data placement and content delivery in p2p networks

pros
1. this draft is easy to follow
2. your core idea of non-metric data placement is reasonable
3. your distributed solution with game theoretic decomposition is similar and closely linked with distributed learning, like clustering based on exploration and exploitation, etc

cons
1. a major concern is that your current draft has no impressive empirical results to support
2. missing strong baselines, some state-of-the-art distributed methods are not tested
3. what's the practical usefulness of this proposed method is unclear

Overall, it's enjoyable to have a read on this draft, it would be glad to recommend towards acceptance, after this work thoroughly be polished based on all comments.

**Questions:**

1. your real-world data source is missing, try to add at least one or two
2. the number of data points or scale in your experiments needs to be verified
3. try to introduce some production data from companies, etc, to better demonstrate the usability of the proposed approach
4. there are related state-of-the-art you may want to compare: Distributed Clustering of Linear Bandits in Peer to Peer Networks, Fast Distributed Bandits for Online Recommendation Systems

**Reviewer Confidence:**

4: The reviewer is certain that the evaluation is correct and very familiar with the relevant literature

**Scope:**

4: The work is relevant to the Web and to the track, and is of broad interest to the community

---

### Official Review · Reviewer_ADmW · 2023-11-21

**Novelty:** 1
**Technical Quality:** 1

**Review:**

This is not my area of expertise, and I am unable to review this submission.
The scores mean nothing and do not reflect the quality of the paper. The system does not allow me to submit it without entering the scores.

**Questions:**

This is not my area of expertise, and I am unable to review this submission.

**Reviewer Confidence:**

1: The reviewer's evaluation is an educated guess

**Scope:**

1: The work is irrelevant to the Web

---

### Official Review · Reviewer_Ugvq · 2023-11-23

**Novelty:** 4
**Technical Quality:** 5

**Review:**

Thank you for submitting your work to WebConf 24. The paper is interesting,  and provide a couple of interesting approximations to solving the non-metric data placement problem. Despite the excellent theoretical approximation of the different proposed algorithms, I left wondering how this can be used in the real-world. I believe that the paper would benefit from an evaluation section of using the devised solutions/algorithms in a real-world setting/problem, highlighting the benefits of using them.

The paper also discusses that the natural Glauber dynamics can collectively result in good suboptimal approximation within a logarithmic factor. It would be great to know how far is that suboptimal approximation is from the optimal solution? Providing a bound would be beneficial.

**Questions:**

- how hard would it be to use a real-world example to highlight the benefit of the proposed algorithms? what about the computational complexity of these algorithms?
- are there any limitations that can arise from translating the theoretical solutions presented in the paper into deployable solutions on the ground?
- how far is the sub-optimal solution from the real optimal one? can you provide at least a bound of how far this can be?

**Reviewer Confidence:**

3: The reviewer is confident but not certain that the evaluation is correct

**Scope:**

3: The work is somewhat relevant to the Web and to the track, and is of narrow interest to a sub-community

---

### Official Review · Reviewer_uzi2 · 2023-11-28

**Novelty:** 4
**Technical Quality:** 4

**Review:**

The paper studies the problem of distributed data placement in distributed settings with non-metric cost. The authors prove that the data placement with non-metric access cost is NP-hard to approximate within a log factor. As a remedy, they propose a novel game theoretic decomposition of the problem that aligns the global objective with agents' selfish behavior. The authors provide a bound on the objective value of any pure Nash equilibrium and additionally using Glauber dynamics provide trade-off between computation cost vs the optimality of the achieved solutions.

S1: The motivation for studying the non-metric version of the problem is natural.

S2: The game theoretic decomposition of the data placement problem is novel.

S3: The work provides the first distributed algorithm for the non-metric version of the data placement problem.


W1: In the absence of any experimental validations, it is difficult to judge the effectiveness of the approach from a practical point of view.

W2: The technical novelty of the work is not clear to me based on the discussions in the related work section. For example, are there prior works on this topic that utilizes Glauber dynamics for finding global optimal solutions?

**Questions:**

W1 and W2.

**Reviewer Confidence:**

3: The reviewer is confident but not certain that the evaluation is correct

**Scope:**

3: The work is somewhat relevant to the Web and to the track, and is of narrow interest to a sub-community

---

### Official Review · Reviewer_JCze · 2023-12-02

**Novelty:** 5
**Technical Quality:** 5

**Review:**

This paper studied the non-metric data placement problem using the game-theoretic decomposition and developed distributed computation algorithms for calculating the optimal solution. This paper made a number of theoretically-grounded contributions, including
(i) NP-hardness of the data placement problem approximation with nonmetric access costs within a factor better than $O(\log(n))$; (2) a game-theoretic decomposition of the data placement problem; and (iii) a bound on the objective value of any pure Nash equilibrium (NE) in terms of the sensitivity to agents’ cache contents. Notations are clearly defined. However, this paper is heavy on notation and very unintuitive. It would be better to improve the presentation by adding motivating examples, tables containing frequently used notations, figurative illustrations, informal descriptions, or empirical results.

**Questions:**

1. Is it possible to move the theoretical proofs to the appendix and use the space for motivating examples, tables containing frequently used notations, figurative illustrations, informal descriptions, or empirical results?
2. How do you justify that the theoretical findings are non-trivial? How do the obtained results compare to previous works?
3. Lemma 3.1 and Lemma .2 in the appendix, Lemma 2.2 and Lemma .3 in the appendix seem to be the same but are named using different numbers.

**Reviewer Confidence:**

3: The reviewer is confident but not certain that the evaluation is correct

**Scope:**

3: The work is somewhat relevant to the Web and to the track, and is of narrow interest to a sub-community

---

### Decision · Program_Chairs · 2024-01-22

**Decision:**

Accept

**Comment:**

# Strengths:
 * Novel Approach: The paper introduces a novel game-theoretic decomposition of the non-metric data placement problem, providing a new perspective on distributed data placement.
 * Theoretical Contributions: Significant theoretical contributions include proving the NP-hardness of the problem, a bound on the objective value of any pure Nash equilibrium, and the polynomial mixing time of the Glauber dynamics for a certain range of noise parameters.
 * Clear Notations: The notations used in the paper are well-defined, aiding in the clarity of the presented concepts.
 * Potential for Real-world Applications: The proposed solution has relevance to practical scenarios like product allocation in distribution networks and P2P data sharing networks.
 # Weaknesses:
 * Lack of Intuitive Presentation: The paper is heavy on notation and could benefit from more intuitive explanations, examples, and empirical results to improve understanding.
 * Absence of Empirical Validation: The lack of experimental results makes it challenging to assess the practical effectiveness of the proposed approach.
 * Comparison with Existing Work: It is unclear how the paper's findings compare to existing work in the field, particularly regarding the use of Glauber dynamics for global optimal solutions.
 * General Presentation and Coherence: The overall flow and coherence of the paper need improvement. Certain sections, like the explanation of models and results, could be better structured.
 # Overall Evaluation:
 The paper offers a theoretically rich approach to solving the non-metric data placement problem in distributed settings, addressing a relevant issue in web caches and content delivery. Its game-theoretic perspective and the use of Glauber dynamics are novel contributions to the field. However, to enhance its impact, the paper would benefit significantly from incorporating practical examples, empirical results, and a more intuitive presentation of the complex theoretical concepts. Comparing the proposed solutions with existing methods and discussing the practical implications of the findings would also add value. With these improvements, the paper has the potential to make a substantial contribution to the understanding and solving of non-metric data placement challenges.